# Global height-resolved methane retrievals from the Infrared Atmospheric Sounding Interferometer (IASI) on MetOp

Richard Siddans<sup>1,2</sup>, Diane Knappett<sup>1,2</sup>, Alison Waterfall<sup>1</sup>, Jane Hurley<sup>1</sup>, Barry Latter<sup>1,2</sup>, Brian Kerridge<sup>1,2</sup>, Hartmut Boesch<sup>2,3</sup>, Robert Parker<sup>2,3</sup>

<sup>1</sup> STFC Rutherford Appleton Laboratory, Chilton, United Kingdom
 <sup>2</sup> National Centre for Earth Observation, United Kingdom
 <sup>3</sup> University of Leicester, United Kingdom

Correspondence to: R. Siddans (richard.siddans@stfc.ac.uk)

- Abstract. This paper describes the global height-resolved methane (CH<sub>4</sub>) retrieval scheme for the Infrared Atmospheric Sounding Interferometer (IASI) on MetOp, developed at the Rutherford Appleton Laboratory (RAL). The scheme is novel in: (a) precisely fitting measured spectra in the 7.9 micron region to allow information to be retrieved on two independent layers centred in the upper and lower troposphere and (b) making specific use of nitrous oxide (N<sub>2</sub>O) spectral features in the same spectral interval to directly retrieve effective cloud parameters to mitigate errors in retrieved methane due to residual
- cloud and other geophysical variables. The scheme has been applied to analyse IASI measurements between 2007 and 2015. Results are compared to model fields from the MACC greenhouse gas inversion and independent measurements from satellite (GOSAT), airborne (HIPPO) and ground (TCCON) sensors. The scheme is shown to be capable of retrieving column average methane with random errors of ~20-40 ppbv on individual soundings. Systematic differences with the other datasets are typically < 10 ppbv regionally, and < 5 ppbv globally. The data has been made publically available via CEDA</p>

(http://dx.doi.org/10.5285/B6A84C73-89F3-48EC-AEE3-592FEF634E9B).

## 1 Introduction

Methane  $(CH_4)$  is one of the most important long-lived greenhouse gases in the atmosphere. Its concentration in the troposphere has increased by a factor of around 2.5 since pre-industrial times, mainly as a result of human activity (IPCC

- 2013). Currently, natural and anthropogenic sources have similar global annual magnitudes. The largest sources are from fossil fuel extraction/use, ruminant livestock, decomposition of waste, rice cultivation (all anthropogenic), wetland emissions, geological sources, termites (natural) and biomass burning (both anthropogenic and natural). Current emission estimates are subject to considerable uncertainties (IPCC 2013, Kirschke 2013), and prediction into the future is even more uncertain. Large amounts of methane are stored in Arctic permafrost and clathrates on the ocean floor: methane release from
- these stores, as a result of warming, would lead to a strong positive feedback on climate.

The atmospheric abundance of methane is determined by surface emissions, balanced by chemical sinks; predominantly via reaction with hydroxl radical. Between 1999 and 2006 there was little growth in the annually-averaged global methane concentration, but since then a significant increase of around 5 ppbv per year has been observed at surface level (Rigby 2008, Sussman 2012). The reasons for this recent behaviour are not yet clear (Nisbet 2014).

- Global observations from nadir-viewing satellite sensors can now provide information to complement and extend that available from surface *in-situ* and remote-sensing measurements to improve our knowledge of the processes controlling the atmospheric distribution of methane, and to monitor its variability on a decadal scale. Methane can be measured remotely using bands in the shortwave infra-red (SWIR) and thermal infra-red (TIR). Observations in the SWIR, such as those from Envisat-SCIAMACHY (Buchwitz 2005, Frankenberg 2010) and GOSAT-TANSO-FTS (Butz 2010, Parker 2011 and
- Yoshida 2013) provide information on column average methane with a vertical sensitivity, in cloud-free conditions over most land surfaces, which is close to uniform throughout the atmospheric column. Such measurements rely on surfacereflected sunlight, so observations are limited to daytime and predominantly over land (ocean reflectance being too low except in sun-glint geometry). TIR measurements of methane are available from spectrometers including Aura-TES (Worden 2012) Aqua-AIRS (Xiong 2008) and MetOp-IASI (Razavi 2009, Crevoisier 2013, Xiong 2013). These complement the
- SWIR observations in regard to both vertical-sensitivity and geographical/temporal coverage. Because TIR spectral signatures depend on thermal contrast between the atmosphere and surface, sensitivity tends to be strong in the mid to upper troposphere and relatively low near the surface. TIR measurements are made over both land and sea, and during day and night. The spatial sampling of IASI is also much greater than that of GOSAT (the only currently operating SWIR methane sensor following the loss of Envisat in 2012): IASI provides 1.3 million soundings per day (with 12 km diameter footprint at
- 20 nadir), over a sufficiently wide swath to provide approximately even sampling of all longitudes (Clerbaux 2009, Eumetsat 2014). IASI is currently flying on both MetOp-A (since 2006) and -B (since 2012); effectively doubling the spatial sampling. Eumetsat plans MetOp-C to take over from -A around 2018, to be followed by IASI Next Generation on the MetOp 2<sup>nd</sup> generation series from 2022-2040; yielding a self-consistent global data set from 2007 onwards. In contrast, GOSAT typically provides of order 1000 soundings per day (10.5 km diameter footprint), over a relatively narrow swath about the 14
- ground tracks (Kuze 2016, Crisp 2012).] The potential for TIR and SWIR observations to be used together to infer nearsurface methane concentrations is studied in (Worden 2015). This paper describes the global height-resolved methane retrieval scheme developed at the Rutherford Appleton Laboratory (RAL). Novel aspects of the scheme include:
- Fitting measured spectra in the 1232-1288 cm<sup>-1</sup> interval to typically ~0.1 K root mean square (RMS) precision, allowing information to be retrieved on two independent height layers centred in the upper and lower troposphere.
- Sophisticated use of nitrous oxide (N<sub>2</sub>O) spectral features in the same interval to estimate effective cloud parameters which mitigate errors in retrieved methane due to residual cloud and other geophysical variables affecting radiative transfer. Although existing schemes (Razavi 2009, Worden 2012) make use of the fact that N<sub>2</sub>O

retrievals from the same spectral range as  $CH_4$  are affected similarly by errors such as residual cloud contamination and temperature errors, their approaches jointly retrieve methane and N<sub>2</sub>O and correct methane *post-hoc*, based on the difference between retrieved N<sub>2</sub>O and its assumed distribution, relying on the fact that tropospheric N<sub>2</sub>O is very close to being uniformly mixed throughout the troposphere with a mixing ratio known to a high degree of accuracy. In the scheme described here, we explicitly model the N<sub>2</sub>O vertical profile spanning the stratosphere, where it varies strongly with height, latitude and season, as well as the troposphere, where it is specified to be uniformly-mixed. The N<sub>2</sub>O spectral signature is instead used to co-retrieve an effective cloud-fraction and height for the fit window, yielding a physically-consistent and more accurate retrieval of methane. An important consequence of modelling the cloud effect directly, compared to the *post-hoc* correction approach using co-retrieved N<sub>2</sub>O, is that the effect of cloud on the vertical sensitivity of the retrieved  $CH_4$  is included in the resulting averaging kernels. We also note that joint retrieval of cloud with trace gases is a fundamental part of the TES algorithm described by Kulawik 2006 and Eldering 2008, however this uses a much wider spectral range (8-15 microns). The specific use of N<sub>2</sub>O absorption lines within the relatively narrow methane fit range, coupled to sophisticated modelling of the N<sub>2</sub>O stratospheric distribution, is key to ensuring that the simplistic representation of cloud in the retrieval is able to effectively represent the impact of cloud on the CH<sub>4</sub> signal.

## 15

20

30

10

5

#### 2 Data Processing Scheme

#### 2.1 **Optimal Estimation**

The scheme is based on the optimal estimation method (OEM, Rodgers 2000), which solves an otherwise under-constrained inverse problem by introducing prior information. This method finds the optimal state-vector  $\boldsymbol{x}$  (which contains the parameters we wish to retrieve) by minimising a cost function:

$$\chi^{2} = (\mathbf{y} - F(\mathbf{x}))^{T} \mathbf{S}_{\mathbf{y}}^{-1} (\mathbf{y} - F(\mathbf{x})) + (\mathbf{a} - \mathbf{x})^{T} \mathbf{S}_{\mathbf{a}}^{-1} (\mathbf{a} - \mathbf{x}), \qquad (1)$$

where y is a vector containing each IASI spectral brightness temperature measurement used by the retrieval;  $S_y$  is a covariance matrix describing the errors on the measurements; F(x) is the forward model (FM), which predicts measurements given x;  $S_a$  is the *a priori* covariance matrix, which describes the assumed errors in the *a priori* estimate of the state, a. As

25 the FM is non-linear with respect to perturbations in methane and other elements of the state-vector, the solution state needs to be found iteratively. In this case we adopt the well-known Levenburg-Marquardt method (summarised in Press 1995), assuming convergence to have occurred when the change in cost-function value is smaller than 1.

#### 2.2 Measurements

IASI (Blumstein 2004) provides spectra at 0.5 cm<sup>-1</sup> apodised resolution, sampled every 0.25 cm<sup>-1</sup>, from 625 to 2760 cm<sup>-1</sup>. Spectra are measured with 4 detectors, each with a circular field of view on the ground (at nadir) of approximately 12 km

diameter, arranged in a 2 x 2 grid within a 50 x 50 km field-of-regard (FOR). IASI scans to provide 30 FORs (120 individual spectra) evenly distributed across a 2200 km wide swath. Our retrieval scheme uses measurements between 1232.25 and 1288.00 cm<sup>-1</sup>, chosen following the work of Razavi (2009) to: (a) minimise errors caused by neglecting line-mixing in the forward model and (b) include channels with relatively clear transmission to the ground, to help constrain co-retrieved cloud parameters and surface temperature. Channels between 1245-1246.75 cm<sup>-1</sup> and 1267-1270 cm<sup>-1</sup> are omitted to avoid problematic spectral features (in the former range attributed to line-mixing effects).

The noise on individual IASI spectra is particularly low in this spectral range (Hilton 2012), with noise-equivalent brightness temperature (NEBT) around 0.07 K (for a reference scene temperature of 280 K), corresponding to a noise equivalent spectral radiance (NESR) of 5.8 nW/cm<sup>2</sup>/cm<sup>-1</sup>/sr. Early retrievals from this range revealed a significant contribution from scene photon noise, so we adopt the following in-house model of the estimated error in each channel:

$$\Delta y_{noise} = \sqrt{h + o\,\bar{I}}\,,\tag{2}$$

where parameters *o* and *h* are constants derived emprically from an analysis of the random component of fit residuals in this range (from an early version of the retrieval scheme) and  $\bar{I}$  is the mean spectral radiance over the complete IASI band. This yields NESR values ranging from 3 to 10 nW/cm<sup>2</sup>/cm<sup>-1</sup>/sr over the typical range of band-averaged measured radiances.

#### 2.3 Forward model

RTTOV version 10 (Matricardi 2009) is the basis of the RAL FM. RTTOV estimates radiances convolved with the IASI spectral response function by use of spectrally-averaged layer transmittances, based on a fixed set of coefficients which weight atmospheric-state-dependent predictors. The RTTOV v10 model is sufficiently fast to enable global processing of the

- IASI mission with modest computational resources. In order to allow interference from the water vapour isotopologue HDO to be modelled adequately, this is handled as an independent variable to the major isotopologue  $H_2^{16}O$ . We have derived coefficients specifically for this spectral range, by running the line-by-line Reference Forward Model (RFM) (Dudhia 2016) using HITRAN 2008 (Rothman 2009) spectroscopic line data for the same set of atmospheric profiles and predictors used in (Matricardi 2009), with the exception that predictors of the type used for CO are used instead to predict HDO transmittances.
- The accuracy of the RTTOV model with the new HDO coefficients is tested by comparing radiances from RTTOV to those calculated directly with the RFM for an independent set of atmospheric profiles. The mean and standard deviation of these differences are generally found to be

The ECMWF ERA-Interim (Dee, 2011) re-analysis is used to define the atmospheric temperature profile and the surface pressure appropriate for each IASI scene. Re-analysis fields are provided at 6 hourly intervals, approximately 0.7 degrees horizontal resolution, on 60 vertical levels. These fields are linearly interpolated to the IASI location and time. Profiles of methane and water vapour (including the isotopologue HDO) are defined by the retrieval state vector (see below), as is the surface (skin) temperature.

5 surface (skin) temperature.

A fundamental distinguishing feature of the RAL scheme is the sophisticated approach developed to specify the  $N_2O$  vertical profile at each IASI measurement location for use in the FM.  $N_2O$  is modelled so that its spectral features can be exploited to co-retrieve two effective cloud parameters, and thereby mitigate related errors on the retrieved methane. In the troposphere,  $N_2O$  exhibits variations with latitude and season smaller than ~0.5% (IPP, 2013); approximately an order of magnitude

- smaller than those of methane. The annual growth rate of  $N_2O$  since IASI became operational in 2007 (around 0.23%/year) has been much more consistent than that of methane in recent decades. Mixing ratios of both  $N_2O$  and methane decrease with height in the stratosphere. In the case of  $N_2O$ , the decrease commences at lower altitude and is steeper than for methane<sup>1</sup>. It is therefore particularly important to accurately model the stratospheric  $N_2O$  profile pertaining to individual IASI observations. The lifetime of  $N_2O$  in the stratosphere is such that spatial and day-to-day variability is controlled by
- dynamics, and can therefore be modelled accurately by exploiting the strong correlation, on potential temperature surfaces, between N<sub>2</sub>O and potential vorticity. This is implemented in the RAL scheme using the seasonal N<sub>2</sub>O climatology derived from the ACE-FTS solar occultation sensor (Jones 2012) which is expressed as a function of equivalent latitude (Lary 1995) and pressure. This is filled below the tropopause with a fixed mixing ratio of 322 ppbv, a representative value for the global mean at the beginning of 2009. The zonal mean field is linearly interpolated to the day of year of a given IASI observation.
- The N<sub>2</sub>O vertical profile at the locations of individual IASI observations is estimated using the local equivalent latitude, derived from potential vorticity and potential temperature given by the ERA-interim re-analysis. The long-term, monotonic growth rate in N<sub>2</sub>O is modelled by scaling each derived profile by the factor: f = 1 + 0.0023d, where d is the number of elapsed days since the beginning of 2009.

Over sea, the RTTOV sea surface emissivity model is used. Over land, the retrieval currently uses the University of

25 Wisconsin surface emissivity database (Seeman 2008). Principal components of an ensemble of measured land surface spectral emissivity spectra are used in conjunction with 0.1 degree latitude/longitude gridded monthly data from MODIS to generate global maps of spectral resolved emissivity.

It is found that fits to measured spectra based on this FM lead to residuals (i.e. differences between observed and modelled spectra) which are small (<0.5 K RMS) but still significant compared to the IASI NEBT (see section 2.2). Column average

mixing ratios from retrieved methane profiles are positively biased compared to independent measurements by approximately 4%, with a systematic height-dependent structure in the profile. A similar bias has been found in TES retrievals which exploit the same spectral range (Worden 2012) and also in MIPAS stratospheric retrievals in limb geometry

<sup>&</sup>lt;sup>1</sup> In the case of  $N_2O$  the decrease with height is due to UV photolysis whereas for methane (CH<sub>4</sub>) it is due to reaction with  $O(^1D)$ .

(Von Clarmann 2009), suggesting a cause related to spectroscopic line parameters. To assess this, retrievals have been carried out substituting RTTOV with the RFM and LBLRTM line-by-line model, with both Hitran 2008 and 2012 line data. LBLRTM includes an approximate treatment of line mixing (Alvaredo 2013). None of these tests led to a significant reduction in the positive bias with respect to independent (TCCON) data. The following approach was therefore taken to empirically address these forward model related issues:

- 1. On input to RTTOV, all methane mixing ratios are multiplied by a factor 1.04 compared to the values in the state vector (i.e. the FM assumes 4% more methane than is ultimately reported as the retrieved value).
- 2. Spectral mean residual patterns derived from applying the retrieval scheme to one day of IASI data over cloud-free ocean in the latitude range of 75° S to 15° S are adopted to represent the mean forward model error and its scanangle dependence. In this version of the scheme, rather than retrieve a height-resolved profile, the methane profile shape is specified, using a similar approach to that adopted to define N<sub>2</sub>O, and a single scale factor retrieved. The resulting mean residual spectrum from the nadir view,  $r_0$ , has negligible correlation with the spectral weighting function for the altitude-independent scaling of the methane profile. The scan-angle dependent component of the FM error is estimated as  $r_1 = r_E r_0$ , where  $r_E$  is the mean residual from both outer edges of the swath.
- 3. To retrieve global methane retrievals, a version of the FM which accounts for the spectral mean residual by adjusting the measurement vector as:

$$y = y_{RTTOV} + f_0 r_0 + f_1 r_1,$$
(3)

where  $\mathbf{y}_{RTTOV}$  is the radiance vector predicted by RTTOV and factors  $f_0$  and  $f_1$  are co-retrieved with methane and other parameters in the state vector, on a scene-by-scene basis.

It is noted that spatial variations in retrieved values of  $f_0$  suggest that the mean residual features are at least partly related to the water vapour distribution.

# 2.4 Selection of IASI observations for processing

The brightness temperature (BT) difference between the IASI observation in a window channel (950 cm<sup>-1</sup>) and that simulated on the basis of ERA-interim, assuming clear-sky conditions, is used to screen out scenes which are strongly affected by

25 cloud. If this difference (observation – simulation) is outside the range of -5 to 15 K, the scene is not processed. Furthermore, only scenes having a BT larger than 240 K in the same channel are currently processed, since (a) the retrieval information content is significantly degraded over very cold surfaces (see below) and (b) convergence is often found to be slow in these conditions, leading to a disproportionate use of computational effort.

The IASI MetOp-A orbits from 29 May 2007 to 17 November 2015 have all been processed with this approach, selecting from the four pixels in each field-of-regard the IASI pixel with the warmest BT at  $950 \text{ cm}^{-1}$ .

# 

#### 2.5 State vector and *a priori* constraint

The state vector for the retrieval scheme consists of 34 elements, as follows:

Methane mixing ratio (in ppmv, relative to dry air) defined on 12 fixed pressure levels corresponding to z\* values of 0, 6, 12, 16, 20, 24, 28, 32, 36, 40, 50, 60 km, where z\* is a simple transformation of pressure, p, (in hPa) to approximate geometric altitude (km):

$$z^* = 16(3 - \log_{10} p) , \qquad (4)$$

The *prior* state is defined to be a fixed value of 1.75 ppmv in the troposphere, broadly representative of southern hemisphere mid-latitudes in 2009, and an annual average height-latitude cross-section in the stratosphere. The zonal mean is calculated by averaging into 5 degree latitude bins two years (September 2008-10) of output from the TOMCAT chemical transport model run into which ACE-FTS observations of long-lived stratospheric tracers have been assimilated (Chipperfield 2002). *A priori* errors are estimated as the root-mean-square combination of the standard deviation of the model methane field about its zonal mean and 10% of the *a priori* value itself. The model standard deviations in the troposphere are around 5%, so *a priori* errors in the troposphere are set to 175 ppb (i.e. 10% of the *prior* value), increasing (in fractional terms) above the tropopause up to peak values of around 50%. The *prior* state and errors are interpolated in latitude to the location of a given IASI observation. To help regularise the retrieval, off-diagonal elements are defined assuming a Gaussian function in the vertical with full-width half-maximum 6 km (in *z*\* units). Here we make a deliberate choice to define a simple and relatively weak constraint to emphasise the vertical resolved information in the IASI measurements. A tighter constraint, which could be justified based on climatological variability, is not necessary to regularise the retrieval and would increase the bias towards the prior (already evident towards high latitudes from the evaluation reported below).

• Natural logarithm of the water vapour (H<sub>2</sub>O) mixing ratio (in ppmv) defined on 16 fixed pressure levels corresponding to z\* values of 0, 1, 2, 3, 4, 5, 6, 8, 10, 12, 16, 20, 30, 40, 50, 60 km. The *a priori* profiles for water vapour are taken from ECMWF analysis, interpolated (linearly) onto these pressure levels. The *a priori* error covariance is intended to represent (conservatively) errors in the analysis and those associated with interpolation to the time and location of an individual IASI observation:

$$S_{a:H20} = S_{bg} + S_{dt} + S_{dx}$$
, (5)

where  $\mathbf{S}_{bg}$  is the ECMWF model background error covariance matrix taken from (Collard 2007);  $\mathbf{S}_{dt}$  is the covariance of differences (considering all profiles in a given day) between ECMWF profiles at one of the 6 hourly analysis times and those at the next analysis time;  $\mathbf{S}_{dx}$  is the analogous covariance of differences between neighbouring spatial grid points. This results in a covariance matrix with relatively large diagonal elements

10

(corresponding to standard deviations of up to 60% peaking in the mid-troposphere), but sufficient vertical correlation to regularise the retrieval.

- Scale factor for HDO, with *a priori* value of 1 and assumed *prior* error of 1. This defines the HDO profile assumed in the FM as follows:
- 5  $r_{HDO}(p) = f_{HDO} f_{std} r_{H2O}(p)$ , (6) where  $f_{HDO}$  is the retrieved factor;  $f_{std}$  is the (fixed) ratio of HDO:H<sub>2</sub><sup>16</sup>O assumed by HITRAN (3.107 × 10<sup>-4</sup>) and  $r_{H2O}(p)$  is the (retrieved) water vapour main isotope mixing ratio profile.
  - Natural logarithm of the (effective) cloud fraction and the associated cloud pressure (hPa). Cloud is modelled in RTTOV as a black body at the given atmospheric level, occupying a given geometric fraction of the scene. The *a priori* cloud fraction is assumed to be 0.01 (before converting to log) and the *a priori* error is assumed to be 10 (which, given the log representation, is a fractional error of 1000% with respect to the prior value of 0.01). The *a priori* cloud pressure is 500 hPa, with error 500 hPa. The logarithm is used as it is found to lead to more stable convergence (non-physical states with negative cloud fraction can otherwise arise during the cost-function minimisation).
- Surface temperature, with *a priori* value taken from ECMWF analysis and assumed *a priori* error of 5 K.
  - The two scale factors for systematic residuals. A priori values are 1 for  $f_0$  and 0 for  $f_1$ . Both are assigned an *a priori* error of 1.

Results from this retrieval scheme are compared below to an earlier version of the scheme, which is identical except that (i) the two cloud parameters are *not* retrieved (scenes passing the initial cloud test are assumed cloud-free) and (ii) the  $N_2O$ 

profile is jointly retrieved, rather than being modelled. The  $N_2O$  prior state and errors are defined in the same way as methane, except that the profiles are scaled, by the same factor at all altitudes, to give a peak mixing ratio in the troposphere of 0.319 ppmv.

## 3 Error analysis and retrieval characterisation

The error covariance of the solution from an optimal estimation retrieval is given by:

$$S_{x} = \left(S_{a}^{-1} + K^{T}S_{y}^{-1}K\right)^{-1}, \tag{7}$$

where *K* is the weighting function matrix which contains the derivatives of the FM with respect to each element of the
30 (solution) state vector. The square-roots of the diagonal elements of this matrix are referred to as the *estimated standard deviation* (ESD) of each element of the state vector.

(8)

The transformation from the retrieved methane mixing ratio profile, x, to the dry-air column averaged mole fraction, c, is expressed as a matrix operation c = Mx, where M contains the weights required to preform the linear operations (i) interpolate the profile defined on the state vector grid to the finer grid used in the radiative transfer model (ii) integrate to give the total column amount (iii) normalise by the total column of air. The ESD of the column average is then given by

5 
$$\Delta c = \sqrt{M S_x M^T}$$
.

The sensitivity of the retrieval to perturbations in the measurement is given by the gain matrix,  $G = S_x K^T S_y^{-1}$ . The sensitivity of the retrieval to perturbations in the true state vector is characterised by the averaging kernel, A = GK.  $S_x$  can be divided into two terms:

$$10 \quad \boldsymbol{S_x} = \, \boldsymbol{S_n} + \, \boldsymbol{S_s} \ , \tag{9}$$

where  $S_n = G S_y G^T$  describes the uncertainity due to measurement errors (characterised by  $S_y$ ) and  $S_s = (I - A)S_a(I - A)^t$  describes the smoothing error, i.e. departure from the true state caused by the tendency of the retrieval towards the imposed *a priori* constraint. As discussed in von Clarrman (2013), the smoothing error only applies to the profile as

- represented on the (rather coarse) retrieval grid. A consequence of this is that the ESD of the total column from equation 8 does not fully capture errors arising from the insensitivity of the retrieval to fine scale perturbations in the vertical profile. Since methane is well-mixed in the troposphere, this issue will often be of little consequence, although there will be a tendency to underestimate total column errors (as well as the column itself) in the vicinity of strong sources where relatively high methane concentrations are present close to the ground. Vertical smoothing errors will be generally overestimated as the
- prior uncertainty on methane exceeds its natural variability. However, the ESD computed in this way corresponds well to the RMS difference between individual IASI measurements and independent TCCON data (see below). Issues relating to the sensitivity of the retrieval to vertical structure and smoothing errors can be addressed using the finescale averaging kernel:

$$A_f = GK_f \tag{10}$$

The weighting function  $K_f$  is distinguished from K in that derivatives may be computed with respect to perturbations on a finer grid than that used for the state vector. The averaging kernel for the sub-column average is  $A_{cf} = MA_f$ . The trace of the averaging kernel A, evaluated using the weighting functions with respect to the retrieved state (A = GK), gives the degrees of freedom for signal (DOFS), which indicates the number of independent pieces of information which can be recovered from the retrieval.

Figure 1 shows averaging kernels for methane, typical of mid-latitude, for nadir observing conditions. The surface temperature is assumed equal to the temperature of the lowest atmospheric layer. The panel on the left shows results assuming an NEBT at 280 K of 0.5 K, while that on the right shows results for 0.1 K (commensurate with the actual IASI