# Peer review of "Global height-resolved methane retrievals from the Infrared Atmospheric Sounding Interferometer (IASI) on MetOp"

_Atmospheric Measurement Techniques, 2016_

## Referee Comment (RC1) · Anonymous Referee #1 · 16 Jan 2017

The authors describe a new methane retrieval product from IASI and show extensive evaluation with model, satellite, ground-based and aircraft retrieval. They highlight the novelty of their approach and make significant effort to compare their data with others, despite different weighting functions. The paper is well written and I would recommend publication, provided the following points are properly addressed:

[Figure]

**Major comments**

- Averaging kernels have been commonly used in the modelling community for more than 10 years, in particular since the first MOPITT products. In this context, there is no point in comparing the satellite data to simple pressure weighted partial columns from a model. The corresponding figures and discussion should be removed, which would allow better focusing the whole discussion.

- In Section 6, it is not clear whether what the authors call "Estimated Standard Deviation" (ESD) is compared to misfits using individual IASI soundings (as suggested p. 1, l. 18 and p. 9, l. 21) or to misfits using averages of IASI soundings (as suggested p. 18, l. 3). The latter would be wrong in the absence of any information about retrieval-to-retrieval error correlations, but the former would require additional information (how is the IASI sounding selected for a given HIPPO profile? – it would be wrong to re-use a given HIPPO profile several times in the statistics).

- The impact of temperature uncertainty is **very** large (p. 10 and Figure 2) if we compare it to the natural variability of the partial column. In contrast to the ESD (as it is computed), it is very likely highly correlated in space and time. This is obviously a major limitation of the IASI information content and it should be much better highlighted and discussed. In p. 10, l. 18, the authors hope that future use of temperature retrievals from IASI will improve the situation, but (i) errors may be correlated between those retrievals and the assimilated spectral samples (at least through RTTOV) and (ii) ERA-Interim already assimilates HIRS and AMSU radiances from Metop (ok, not from IASI, but at least collocated temperature information is used).

- Section 4 lacks explanation. It suggests that total and partial columns are compared together, which would be wrong, in contrast to what is done later with Eq.

[Figure]

(13). The quality of the retrieved cloud parameters is also not discussed, even though these play a key role in the retrievals. The use of cloud observations from AVHRR instead of these ones suggests, maybe erroneously, that it is quite poor.

- Section 5.2: the authors have chosen to use a GOSAT "proxy" product even though it only provides the methane total column. However, "full-physics" products (e.g., available from the same data providers and co-authors) provide methane profiles: they would be much more appropriate and would (in conjunction with proper use of the averaging kernels – Rodgers and Connor 2003) avoid the not-so-clean trick of Eq. (13).

**Minor comments**

- There are a few typos that should be removed (e.g., p. 4 l. 13, p. 18 l.13 and l. 22).

- The format of the references is not unified.

- Methane units in all figures should be ppb rather than ppm.

- P. 1, l. 18: it should be said that the column is partial.

- p. 1, l. 25-28: the list includes both processes (e.g., fossil fuel use) and explicit sources (e.g., wetland emissions). The format should be harmonized.

- P. 2, l. 14: to be fair, the 2009 paper from LMD should be mentioned (Crevoisier et al. 2009). Actually, since that IASI methane product line has been well established, a few words to explain how the design of this product differs from the new one presented here would be good.

- P. 2, l. 18: Sentinel 5P is about to be launched and could usefully be mentioned.

- P. 3, l. 24: "error" is missing before "covariance".

- P. 4, l. 1: km$^2$ rather than km.

- P. 4, l. 29: because of the large number of spectral samples, this way of doing does not make sense without accounting for the correlated errors (see also Stewart et al., 2014).

- P. 5, l. 28: it is not clear what the FM uses in input.

- P. 6, l. 6-7: the empiricism of this 4% (70 ppb !!!!) correction in the methane amount is quite perturbing.

- P. 7, l. 11: "statistics" is missing before "are estimated".

- P. 20, l. 5-8: the last sentences of the paper are nearly the same as those of Crevoisier et al. (2013) that were actually a simple reformulation of the last lines of their 2009 paper. Either this idea is trivial, and it should be removed, or it would be fair to quote the previous papers.

**References**

Crevoisier, C., Nobileau, D., Fiore, A. M., Armante, R., Chédin, A., and Scott, N. A.: Tropospheric methane in the tropics – first year from IASI hyperspectral infrared observations, Atmos. Chem. Phys., 9, 6337-6350, doi:10.5194/acp-9-6337-2009, 2009.

Crevoisier, C. D. Nobileau, R. Armante, L. Crépeau, T. Machida, Y. Sawa, H. Matsueda, T. Schuck, T. Thonat, J. Pernin, N.A. Scott, and A. Chédin: The 2007-2011 evolution of tropical methane in the mid-troposphere as seen from space by MetOp-A/IASI. Atmos. Chem. Phys., 13, 4279-4289, doi:10.5194/acp-13-4279-2013, 2013.

Rodgers, C. D., and B. J. Connor (2003), Intercomparison of remote sounding instruments, J. Geophys. Res., 108, 4116, doi:10.1029/2002JD002299, D3.

Stewart, L. M., Dance, S. L., Nichols, N. K., Eyre, J. R. and Cameron, J. (2014), Estimating interchannel observation-error correlations for IASI radiance data in the Met Office system†. Q.J.R. Meteorol. Soc., 140: 1236–1244. doi:10.1002/qj.2211

---

## Referee Comment (RC2) · Anonymous Referee #2 · 20 Jan 2017

General comments:

The authors have applied height-resolved methane retrievals to IASI using a modified N2O correction scheme. Given the global coverage and long-time series for IASI, improvements in IASI results will lead to significant scientific advancements.

The abstract states that the scheme is novel in the use of "precisely fitting measured spectra in the 7.9 micron region to allow information to be retrieved on two independent layers centred in the upper and lower troposphere" and "making specific use of nitrous oxide (N2O) spectral features in the same spectral interval to directly retrieve effective cloud parameters to mitigate errors in retrieved methane due to residual cloud and other geophysical variables" were both previously done by Worden et al. (2012). The

statement that this is novel should be removed. The authors could state that they have developed a novel variant of N2O correction.

The abstract and conclusion gives only errors for the column. The abstract and conclusions should give the errors for the new vertically resolved quantities. In the conclusions, if errors are also given for column CH4, these should be compared to current IASI CH4 products.

P8 L20. The article states that the averaging kernel and predicted errors are more accurate in the new scheme versus previous methods where N2O is retrieved and then used to correct CH4. To make this claim, comparisons of results, averaging kernels, and predicted errors need to be shown between the method where N2O is fixed versus retrieved, at least for a few cases. The results where N2O is retrieved must include the retrieval of cloud parameters to make it a fair comparison, as the previous scheme this tries to improve on, e.g. Worden et al. (2012), retrieves cloud parameters jointly with N2O and CH4.

The title of the article is "Global height-resolved methane retrievals..." and yet most of the results shown are column results. Comparisons to TCCON and GOSAT are of limited interest to this paper because they are not vertically resolved. The only reason to compare the total column is if the new retrieval improves over the current IASI column. If this is the case, both new and current IASI CH4 results should be shown.

HIPPO is primarily over ocean. The authors need to validate the height-resolved CH4 over land.

Specific comments:

P.2 Line 18. The word "also" is confusing and should be taken out.

P2 L32 "Sophisticated use" is not a useful description of what the authors are doing. I would reword this to something like, "Use of an fixed N2O volume mixing ratio values,

based on ACE climatology described in Section XX, to retrieve cloud parameters in the CH4 spectral region."

P5 L16. The description of the N2O a priori definition is sufficiently important to the paper that it should be in its own sub-section so that readers can easily find it. It should not be under the heading "forward model".

P3 L13 "coupled to sophisticated modelling...". The use of the word "sophisticated" is not useful. Something like, "coupled to modeling of N2O based on the ACE climatology, parameterized by latitude, pressure, and date, as described in Section ##." Please remove all other instances of the word "sophisticated".

P3 L12 "described by Kulawik 2006 and Eldering 2008, however this uses a much wider spectral range (8-15 microns)" It's not obvious from the above cited papers, but although the cloud parameters are initially estimated using 8-15 microns, cloud parameters are then retrieved in the windows used for each retrieved species. For methane, the windows are 7.5-8.4 microns.

P3 L6-15. It appears N2O is set and then not retrieved. Could you state this explicitly? It's currently confusing to the reader what you are doing.

P5 L29. "Column average mixing ratios from retrieved methane profiles are positively biased compared to independent measurements by approximately 4%, with a systematic height-dependent structure in the profile." It needs to be clarified what is being compared to and a citation or reference to a later section needs to be given.

P5 L31. I'd cite the Alvarado, 2015 paper for TES biases.

P5 L31. The TES bias in Alvarado, 2015 is twice as much in the upper troposphere as lower. MIPAS does not see into the lower Troposphere. If IASI sees a 4% bias in the lower Troposphere and no bias in the upper Troposphere, this result is somewhat different from previous results.

P9 L30. Since IASI columns are being compared to GOSAT and TCCON, it is important

to show the column averaging kernel and compare to that of GOSAT and TCCON (e.g. see Boesch et al. (2011), figure 13). The values in Fig 1 of 0.15 or 0.2 impossible to compare the sensitivity of the column averaging kernel.

P10 L 22 and Figure 4. Please define the DOF's for the column averaging kernel. I don't think a single parameter quantity, e.g column amount, can have degrees of freedom above 1, by definition. Can you show the sum of the trace of the averaging kernel for the CH4 profile, and/or the DOF for the lower and upper partial columns for CH4.

P10 L18 As discussed in general comments, to support the assertion that you have height resolved methane retrievals, in addition to the 6-12 km results, 0-6 km or 3-6 km comparisons to MACC need to be shown.

P18 L16. The HIPPO comparisons in Figs 14-15 of 0-6, 6-12 and total column are good. The biases are summarized in the text. Please also mention the standard deviation in the text, abstract, and conclusions.

Conclusions. The conclusions only discuss results for column CH4. As the paper is on height-resolved methane retrievals, the discussion needs to emphasize results for height-resolved quantities of 0-6 km and 6-12 km.

Figure 6. Show MACC with the IASI averaging kernel (Eq. 12) for day and night. In particular, I'd want to see that the day/night differences seen in IASI are due to sensitivity not variations in CH4.

Figure 9-10. Show comparisons for 0-6 km also.

Figures 11-14. These figures are column-based and do not support the validation of height-resolved methane retrievals. If the assertion is that the full-column retrievals are also improved, results should be compared to current IASI CH4 retrievals. Otherwise results should focus on height-resolved validation.

[Figure]

---

## Author Comment (AC1) · 26 Jul 2017

**Referee comments in bold**
Responses in normal type

**Major comments**

**• Averaging kernels have been commonly used in the modelling community for more than 10 years, in particular since the first MOPITT products. In this context, there is no point in comparing the satellite data to simple pressure weighted partial columns from a model. The corresponding figures and discussion should be removed, which would allow better focusing the whole discussion.**

The importance of using averaging kernels (AKs) is clearly understood, and they are used throughout the paper to allow comparisons with independent observational data and models on a like-with-like basis, taking into account retrieval sensitivity to vertical structure and prior information. However, if the retrieval of a given vertical layer is dominated by prior information or is mostly sensitive to layers above, comparisons with AKs may show excellent agreement, despite the lack of retrieval information in the lower layer. Comparisons *without* application of averaging kernels indicate how well the retrieval observes geographical variability *within* each specified layer. We therefore consider it necessary and informative to show these comparisons as well.

**• In Section 6, it is not clear whether what the authors call "Estimated Standard Deviation" (ESD) is compared to misfits using individual IASI soundings (as suggested p. 1, l. 18 and p. 9, l. 21) or to misfits using averages of IASI soundings (as suggested p. 18, l. 3). The latter would be wrong in the absence of any information about retrieval-to-retrieval error correlations, but the former would require additional information (how is the IASI sounding selected for a given HIPPO profile? – it would be wrong to re-use a given HIPPO profile several times in the statistics).**

It is the former. As described, IASI soundings are selected for each HIPPO profile (all within 200km and 6 hours). These matches are the basis for computing statistics. HIPPO profiles are not reused. This is clarified further in the revised text.

**• The impact of temperature uncertainty is very large (p. 10 and Figure 2) if we compare it to the natural variability of the partial column. In contrast to the ESD (as it is computed), it is very likely highly correlated in space and time. This is obviously a major limitation of the IASI information content and it should be much better highlighted and discussed. In p. 10, l. 18, the authors hope that future use of temperature retrievals from IASI will improve the situation, but (i) errors may be correlated between those retrievals and the assimilated spectral samples (at least through RTTOV) and (ii) ERA-Interim already assimilates HIRS and AMSU radiances from Metop (ok, not from IASI, but at least collocated temperature information is used).**

More discussion on this point has been added to section 2 of the paper.

**• Section 4 lacks explanation. It suggests that total and partial columns are compared together, which would be wrong, in contrast to what is done later with Eq. (13). The quality of the retrieved cloud parameters is also not discussed, even though these play a key role in the retrievals. The use of cloud observations from AVHRR instead of these ones suggests, maybe erroneously, that it is quite poor.**

A footnote is added to explain why using AKs is not necessary for the analysis reported in section 4. We deliberately do not dwell on the physical interpretation of the cloud parameters; they are *effective* parameters intended only to account for cloud (and other) effects on the CH4 retrieval in the 7.9 micron band. They are not expected to compare well with AVHRR parameters (especially not visible optical depth). We consider it sufficient to demonstrate that they clearly do mitigate cloud-related errors on CH4 (as demonstrated in section 4).

**• Section 5.2: the authors have chosen to use a GOSAT "proxy" product even though it only provides the methane total column. However, "full-physics" products (e.g., available from the same data providers and co-authors) provide methane profiles: they would be much more appropriate and would (in conjunction with proper use of the averaging kernels – Rodgers and Connor 2003) avoid the not-so-clean trick of Eq. (13)**.

Even though profiles may be reported, there remains only a single degree of freedom in the SWIR methane retrieval (full-physics or not), so these are no more amenable to a Rodgers and Connors approach than the proxy results. Results from the proxy method have also been demonstrated to be preferable in certain respects to those from the full-physics method, and so were selected for comparison with IASI here.

**Minor comments**

**• There are a few typos that should be removed (e.g., p. 4 l. 13, p. 18 l.13 and l.22).**

We cannot locate these typos.

**• The format of the references is not unified.**

We await editorial guidance on this point.

**• Methane units in all figures should be ppb rather than ppm.**

This is a matter of personal preference rather than convention, and we prefer not to change; ppb and ppm are both used mainly for clarity in plots. While absolute values are more compact when written in ppm (e.g. 1.8 vs 1800) differences derived from comparisons are typically small and often clearer if stated in ppb (10 ppb vs 0.01 ppm).

**• P. 1, l. 18: it should be said that the column is partial.**

The sentence refers to the inferred column average. We have added a clause to the sentence to make it clear that this should not be interpreted without considering the vertical sensitivity.

**• p. 1, l. 25-28: the list includes both processes (e.g., fossil fuel use) and explicit sources (e.g., wetland emissions). The format should be harmonized.**

We have changed the word "sources" in the sentence to "contributions".

**• P. 2, l. 14: to be fair, the 2009 paper from LMD should be mentioned (Crevoisier et al. 2009). Actually, since that IASI methane product line has been well established, a few words to explain how the design of this product differs from the new one presented here would be good.**

Reference to the 2009 paper has been added, together with a paragraph outlining key differences between the RAL and Crevoisier schemes.

**• P. 2, l. 18: Sentinel 5P is about to be launched and could usefully be mentioned.**

Mention has been added.

**• P. 3, l. 24: "error" is missing before "covariance".**

It is clarified later in the sentence that the covariance describes the errors of the measurements.

**• P. 4, l. 1: $km_2$ rather than km.**

This has been corrected.

**• P. 4, l. 29: because of the large number of spectral samples, this way of doing does not make sense without accounting for the correlated errors (see also Stewart et al., 2014).**

A footnote has been added to clarify that error correlations in the modelled RTTOV-related error are unimportant.

**• P. 5, l. 28: it is not clear what the FM uses in input.**

A footnote has been added to clarify that the FM takes input as for the standard retrieval scheme (except without the derived residual patterns).

**• P. 6, l. 6-7: the empiricism of this 4% (70 ppb !!!!) correction in the methane amount is quite perturbing.**

We have expanded slightly on this point in the text.

**• P. 7, l. 11: "statistics" is missing before "are estimated".**

This has been added.

**• P. 20, l. 5-8: the last sentences of the paper are nearly the same as those of Crevoisier et al. (2013) that were actually a simple reformulation of the last lines of their 2009 paper. Either this idea is trivial, and it should be removed, or it would be fair to quote the previous papers.**

The paragraph was drafted wholly independently of the Crevoisier paper so we were unaware of any similarity. The plans to launch Metop-C and Metop-SG are well established, however, here we make specific reference to "global, height-resolved" methane distributions which those missions are capable of providing, and which is a *new* capability with respect to the Crevoisier paper.

---

## Author Comment (AC2) · 26 Jul 2017

**Referee comments in bold.**
Responses in normal type.

**The abstract states that the scheme is novel in the use of "precisely fitting measured spectra in the 7.9 micron region to allow information to be retrieved on two independent layers centred in the upper and lower troposphere" and "making specific use of nitrous oxide (N2O) spectral features in the same spectral interval to directly retrieve effective cloud parameters to mitigate errors in retrieved methane due to residual cloud and other geophysical variables" were both previously done by Worden et al. (2012). The statement that this is novel should be removed. The authors could state that they have developed a novel variant of N2O correction.**

The abstract has been re-phrased accordingly.

**The abstract and conclusion gives only errors for the column. The abstract and conclusions should give the errors for the new vertically resolved quantities. In the conclusions, if errors are also given for column CH4, these should be compared to current IASI CH4 products.**

Errors for vertically resolved quantities are now added to abstract and conclusions.

Crevoisier et al. (2009) estimated precision on methane retrieved by their neural net scheme in the mid/upper tropospheric layer to be 16 ppbv for a 5 × 5 deg spatial resolution on a monthly time scale. Comparison with individual soundings with our scheme on a like-for-like basis could be informative, however, this was beyond the scope of our paper.

**P8 L20. The article states that the averaging kernel and predicted errors are more accurate in the new scheme versus previous methods where N2O is retrieved and then used to correct CH4. To make this claim, comparisons of results, averaging kernels, and predicted errors need to be shown between the method where N2O is fixed versus retrieved, at least for a few cases. The results where N2O is retrieved must include the retrieval of cloud parameters to make it a fair comparison, as the previous scheme this tries to improve on, e.g. Worden et al. (2012), retrieves cloud parameters jointly with N2O and CH4.**

Since we have not quantitatively tested the benefit of the improvement in AKs from our cloud-modelling approach, the text in the introduction (2$^{nd}$ bullet) has been modified.

**The title of the article is "Global height-resolved methane retrievals..." and yet most of the results shown are column results. Comparisons to TCCON and GOSAT are of limited interest to this paper because they are not vertically resolved. The only reason to compare the total column is if the new retrieval improves over the current IASI column.**

The emphasis on column average comparisons is driven by the fact that almost all available independent observational data is of that type. In particular, TCCON provides the most extensive and well-established correlative data set with which to assess the IASI retrievals. We therefore consider it important to compare with TCCON and column average data from GOSAT. Nevertheless, we do take the point, and now include additional results (new figures 7 and 11) comparing the IASI 0-6km layer average with MACC (the only data-set other than HIPPO which allows height-resolved comparisons).

In addition, ESDs and averaging kernels are now presented for the two layers in figures 3 and 4.

**If this is the case, both new and current IASI CH4 results should be shown.**

As stated above, comparison with other IASI methane datasets was not feasible and so beyond the scope of this paper.

**HIPPO is primarily over ocean. The authors need to validate the height-resolved CH4 over land.**

The theoretical performance of our scheme is characterised in the paper over land and sea. We have used the height-resolved observational data from HIPPO which is available. The MACC comparisons have performed over both land and sea, and the TCCON comparisons are predominantly over land. We are not aware of another observational data set against which height-resolved IASI data over land could have been extensively compared.

**Specific comments:**

**P.2 Line 18. The word "also" is confusing and should be taken out.**

Done.

**P2 L32 "Sophisticated use" is not a useful description of what the authors are doing. I would reword this to something like, "Use of an fixed N2O volume mixing ratio values, based on ACE climatology described in Section XX, to retrieve cloud parameters in the CH4 spectral region."**

We believe the wording of Section 2.4 to now be quite clear.

**P5 L16. The description of the N2O a priori definition is sufficiently important to the paper that it should be in its own sub-section so that readers can easily find it. It should not be under the heading "forward model".**

This has been moved to a separate section.

**P3 L13 "coupled to sophisticated modelling...". The use of the word "sophisticated" is not useful. Something like, "coupled to modeling of N2O based on the ACE climatology, parameterized by latitude, pressure, and date, as described in Section ##." Please remove all other instances of the word "sophisticated".**

The sentence has been re-phrased and the word "sophisticated" is also removed elsewhere.

**P3 L12 "described by Kulawik 2006 and Eldering 2008, however this uses a much wider spectral range (8-15 microns)" It's not obvious from the above cited papers, but although the cloud parameters are initially estimated using 8-15 microns, cloud parameters are then retrieved in the windows used for each retrieved species. For methane, the windows are 7.5-8.4 microns.**

This statement in our paper has now been modified to specify that Worden 2012 includes cloud parameters but jointly fits N2O. It is difficult to see where the information on cloud comes from in their approach. We assume it to be well constrained by their broad-band retrieval, however this is not clear from their papers.

**P3 L6-15. It appears N2O is set and then not retrieved. Could you state this explicitly?**
**It's currently confusing to the reader what you are doing.**

This is now stated explicitly.

**P5 L29. "Column average mixing ratios from retrieved methane profiles are positively biased compared to independent measurements by approximately 4%, with a systematic height-dependent structure in the profile." It needs to be clarified what is being compared to and a citation or reference to a later section needs to be given.**

Clarification has now been provided in a footnote.

**P5 L31. I'd cite the Alvarado, 2015 paper for TES biases.**

Citation now added.

**P5 L31. The TES bias in Alvarado, 2015 is twice as much in the upper troposphere as lower. MIPAS does not see into the lower Troposphere. If IASI sees a 4% bias in the lower Troposphere and no bias in the upper Troposphere, this result is somewhat different from previous results.**

We do not wish to contend that the biases are identical, but only to point out that other schemes have been found to exhibit comparable biases, which might point towards spectroscopic issues and possibly thereby motivate future work in that area. The statement has been modified to avoid an impression that the biases are entirely consistent.

**P9 L30. Since IASI columns are being compared to GOSAT and TCCON, it is important to show the column averaging kernel and compare to that of GOSAT and TCCON (e.g.see Boesch et al. (2011), figure 13). The values in Fig 1 of 0.15 or 0.2 impossible to compare the sensitivity of the column averaging kernel.**
**P10 L 22 and Figure 4.**

The averaging kernel for IASI column average methane is now shown in figures 1 and 3. In figure 1 the values of the kernel are divided by 10 (so as to fit on the same range) – this was not sufficiently clear before so we have now added this to the figure caption. Apart from this scaling, the kernel should be directly comparable to that of Boesch et al. 2011. We do not consider it necessary to present their kernels here, however, since whether or not they are used (via eqn 13) makes very little difference to our IASI comparisons to GOSAT or TCCON, so the detailed GOSAT/TCCON kernel shapes are of secondary interest, and they can be seen in the cited reference.

**Please define the DOF's for the column averaging kernel.**

We have presented DOFS for the retrieval state vector. If reformulated to retrieve a methane column average mixing ratio instead of a height-resolved profile, the corresponding DOFS would be unity, so we are not clear what the Referee intends by this request.

**I don't think a single parameter quantity, e.g column amount, can have degrees of freedom above 1, by definition. Can you show the sum of the trace of the averaging kernel for the CH4 profile, and/or the DOF for the lower and upper partial columns for CH4.**

DOFS for the profile are given in figure 1, these are determined from the trace of the averaging kernel. The caption has been extended to clarify this. Layer average mixing ratios are computed from the retrieved profile, as described in Section 3. Please see response to previous point.

**P10 L18 As discussed in general comments, to support the assertion that you have height resolved methane retrievals, in addition to the 6-12 km results, 0-6 km or 3-6 km comparisons to MACC need to be shown.**

These are now included.

**P18 L16. The HIPPO comparisons in Figs 14-15 of 0-6, 6-12 and total column are good. The biases are summarized in the text. Please also mention the standard deviation in the text, abstract, and conclusions.**

Standard deviations and their relation to ESD were already discussed in relation to figure 15 (now figure 17); to which we have added some more quantitative values. The extent to which these results support the estimated random errors on the layer averages is now

mentioned in the conclusions. We do not consider that a specific point in the abstract is necessary.

**Conclusions. The conclusions only discuss results for column CH4. As the paper is on height-resolved methane retrievals, the discussion needs to emphasize results for height-resolved quantities of 0-6 km and 6-12 km.**

Text has been added accordingly.

**Figure 6. Show MACC with the IASI averaging kernel (Eq. 12) for day and night. In particular, I'd want to see that the day/night differences seen in IASI are due to sensitivity not variations in CH4.**

Day – night variations in retrieved CH4 cannot be tested in this way because the MACC data are daily average methane values (as mentioned in the paper), so the differences shown specifically relate to IASI day-night differences in sampling/sensitivity; to assess diurnal variation in methane, a different model would need to be introduced.

**Figure 9-10. Show comparisons for 0-6 km also.**

This has been done (new figure 11).

**Figures 11-14. These figures are column-based and do not support the validation of height-resolved methane retrievals. If the assertion is that the full-column retrievals are also improved, results should be compared to current IASI CH4 retrievals. Otherwise results should focus on height-resolved validation.**

As pointed out above, we consider comparison to TCCON to be important and this can only be done via column average comparisons. Our response to the Referee's earlier comment applies here:

The emphasis on column average comparisons is driven by the fact that almost all available independent observational data is of that type. In particular, TCCON provides the most extensive and well-established correlative data set which to assess the IASI retrievals. We therefore consider it important to compare with TCCON and column average data from GOSAT. Nevertheless, we do take the point, and now include additional results (new figures 7 and 11) comparing the IASI 0-6km layer with MACC (the only data-set other than HIPPO which allows height-resolved comparisons). In addition, ESDs and averaging kernels are now presented for the two layers in figures 3 and 4.